# Post COVID-19 vaccination side effects and associated factors among vaccinated health care providers in Oromia region, Ethiopia in 2021

**Mesfin Tafa Segni** [1]*, **Hailu Fekadu Demissie**[1], **Muhammedawel Kaso Adem**[1], **Adem Kedir Geleto**[2], **Mesfin Wubishet Kelkile**[3‡], **Birhanu Kenate Sori**[4], **Melese Lemmi Heyi**[4], **Dhabesa Gobena Iticha**[4], **Gemechu Shumi Bejiga**[4], **Abera Botere Guddisa**[4], **Yadeta Ayana Sima**[4‡], **Lemessa Tadesse Amente**[4‡], **Dereje Abdena Bayisa**[4‡], **Mengistu Bekele Hurisa**[4‡], **Tesfaye Kebebew Jiru**[4‡]

**1** Department of Public Health, College of Health Science, Arsi University, Asella, Ethiopia, **2** Department of Agricultural Economics, College of Agriculture and Environmental Science, Arsi University, Asella, Ethiopia, **3** Arsi Zone Health Department, Oromia Regional Health Bureau, Asella, Ethiopia, **4** Oromia regional Health Bureau, Addis Ababa, Ethiopia

☯ These authors contributed equally to this work.
‡ These authors also contributed equally to this work.
* mesfintafa2011@gmail.com

**Data Availability Statement:** The raw data stored in SPSS software is attached as Supporting information.

## Abstract

### Background

Severe Acute Respiratory Syndrome (SARS COV-2) known as COVID-19 since its outbreak in 2019, more than 375 and 5.6 million were infected and dead, respectively. Its influence in all disciplines stimulated different industries to work day to night relentlessly to develop safe and effective vaccines to reduce the catastrophic effect of the disease. With the increasing number of people globally who have been vaccinated, the reports on possible adverse events have grown and gained great public attention. This study aims to determine post-COVID-19 vaccination adverse effects and associated factors among vaccinated Health care providers in the Oromia region, Ethiopia in 2021.

### Methods

A cross-sectional study was conducted among 912 health care workers working in government hospitals in the central Oromia region from November 20 to December 15/2021. Respondents absent from work due to different reasons were excluded during the interview. The outcome variable was COVID-19 side effects (response as Yes/No). A descriptive analysis displayed findings in the form of the frequencies and percentages, and logistic regression was employed to see the association of different variables with side effects experienced.

### Result

Overall, 92.1% of the participants experienced side effects either in 1st or 2nd doses of post-COVID-19 vaccination; 84.0% and (71.5%) of participants experienced at least one

**Funding:** Oromia Regional Health Bureau has funded this project for the data collectors perdium with agreement number BEFO/ABFH-D/1543. The funding has no right in the design, analysis and interpretation of the data and in writing the manuscript.

**Competing interests:** The authors have declared that no competing interests exist.

**Abbreviations:** ArsU, Arsi University; CI, Confidence interval; COVID 19, Coronavirus Disease 19; OR, Odds Ratio; SARS, Severe Acute Respiratory infections; SDGs, Sustainable Development Goals; WHO, World Health Organization.

side effect in the 1st and 2nd dose of the vaccines, respectively. COVID-19 infection preventive protocols like keeping distance, hand wash using soap, wearing mask and using sanitizer were decreased post vaccination. About 74.3% of the respondents were worried about the adverse effects of the COVID-19 vaccine they received. The majority (80.2%) of the respondent felt fear while receiving the vaccine and 22.5% of the respondents suspect the effectiveness of the vaccine they took. About 14.8% of the vaccinated Health workers were infected by COVID-19 post-vaccination. Engaging in moderate physical activity and feeling fear when vaccinated were the independent factors associated with reported side effects of post-COVID-19 vaccination using multiple logistic regression. Respondents who did not engage in physical activity were 7.54 fold more likely to develop post-COVID-19 vaccination side effects compared to those who involved at least moderate-intensity physical activity [AOR = 7.54, 95% CI;2.46,23.12]. The odds of experiencing side effects among the respondents who felt fear when vaccinated were 10.73 times compared not felt fear (AOR = 10.73, 95% CI; 2.47,46.64), and similarly, those who felt little fear were 4.28 times more likely to experience side effects(AOR = 4.28, 95% CI; 1.28, 14.39).

## Conclusion

Significant numbers of the respondents experienced side effects post COVID-19 vaccination. It is recommended to provide pre-awareness about the side effects to reduce observed anxiety related to the vaccine. It is also important to plan monitoring and evaluation of the post-vaccine effect using standard longitudinal study designs to measure the effects directly.

## Introduction

Severe Acute Respiratory Syndrome (SARS COV-2) known as COVID-19 latter, since its outbreak in December 2019 in China Hubei province more than 5.6 million lives were lost and caused about 375million morbidity [1]. All sectors are badly affected by the pandemic. COVID 19 is worrisome in countries with low-capacity and humanitarian settings ill-equipped to cope with COVID-19 due to weak health systems and workforces that are heavily reliant on the support of the donors [2–4].

The negative impacts of COVID-19 on social, economic and political in the globe have stimulated different agencies industries to work day to night relentlessly to develop safe and effective vaccines urgently to control the spread of the pandemic [5–10]. In the late of 2020, there were more than 214 vaccines were developed to combat this pandemic, of these 50 have progressed to human clinical trials and some are being given to individuals [8–10]. Nine vaccines have completed phase III clinical trials and have approved by the World Health Organization(WHO). Three from US; Ad26.COV2.s(Johnson & Johnson)), NVX-CoV2373 (Novavax) and mRNA-1273(Moderna-US), three from China CoronaVac (Sinovac Biotech), BBIBP-CorV(Sinopharm(Beijing)) and ConvideciaTM(CanSino Bio); and the rest three from Germany- BNT162b2(Pfizer/BioNTech), Russian- Sputnik V (Gamaleya) and from United Kingdom- AZD1222 (AstraZeneca) [11–17], this approved drug fall in three types; inactivated vaccine, adenovirus-vector vaccine and nucleic acid vaccine [18].

Moreover, the rapid development of vaccines casts doubt on safety. Previously, the rapid developments of the vaccines have been linked to different adverse issues [19,20]. For example, the swine flu vaccine increased the risk of Guillain-Barre syndrome [21].

Billions of doses of vaccines have been administered to adults around the world during the COVID-19 pandemic. With the increasing number of people globally who have been vaccinated, the reports on possible adverse events have grown and gained great public attention [22].

Different studies indicated that vaccinated individuals showed variable adverse effects. The adverse effect varies with the types of vaccines received among population groups. In a study in Saudi Arabia, the reported side effects associated after receiving Oxford-AstraZeneca and Pfizer-BioNTech COVID-19 vaccines were 60% and 68.5%, respectively; and after 1st dose of ChAdOx1-S vaccine was 34.7% [23–25]. In a study in Poland, a small number of post-vaccination reactions were reported. Having earlier suffered from COVID-19 had an impact on the occurrence of more severe side effects after the first dose of the COVID-19 vaccine [26]. A study in a French university hospital among those who received the first dose of Pfizer-BioNTech COVID-19 vaccine, 74% of patients reported at least one side effect. Among participants with a history of COVID-19, 95% reported at least one adverse event versus 70% in naive patients [27]. A study in Slovakia among those receiving the BNT162b2 vaccine, 91.6% reported at least one side effect [28]. In a study in Vietnam; who received at least one dose of AZD1222, 96.1% reported adverse effects [29]. In a study in the USA, 64.5% received the BNT162b2 mRNA vaccine and reported at least one or more symptoms post-vaccination [30]. Post-vaccination infection was also reported by some studies; 0.5% in Saudi [25] and 0.54% in Israel [26].

Ethiopia has been providing vaccination to the health workforce and prioritized population groups including officials since February 2021. Nevertheless, post-COVID-19 vaccine effect of the vaccine is not been studied so far in Oromia region as well in Ethiopia as far as our knowledge goes until this study was conducted. So the aim of this study is to determine post-COVID-19 vaccination adverse effects among health workers in the Oromia region, in Ethiopia in 2021.

## Methods

### Study area and period

The study was conducted in selected facilities (Asella Referral and Teaching Hospitals, Bokoji Hospital, Adama Hospital and Medical College, Mojo Hospital, Bishoftu Hospital, and Shashemene Referral Hospital) in the central Oromia region, Ethiopia from November to December 2021. The region is the largest and most populous region of the 11 regional states found in Ethiopia. Administratively, the region is divided into 21 Zones and 19 administrative towns, 317 districts, and 7011 kebeles (the smallest administrative unit in Ethiopia). The area of the region is 363 399.8 km$^2$, which accounts for 32% of the land size of the country. Based on the Population and Housing Census in 2007, the region's population is projected to be around 38 million by 2020, accounting for nearly 35% of the Ethiopian population; 87.7% are rural residents [31]. There are 1,421 Health Centers, 110 Hospitals, 7090 Health Posts, 10 Blood Banks, 3 Regional Laboratories, as per attached PPT. Regarding the region health workforce there are 77,999 staffs (52,528(67.8%) health professionals, and 25472(32.7%) supportive staffs).

### Study design

A facility-based cross-sectional study was used among vaccinated health care workers in selected hospitals in the central Oromia region, Ethiopia from November to December 2021.

**Source population**: All health care providers who received COVID-19 at least one vaccine dose in the selected health facilities in the Oromia region.

**Inclusion criteria**: All health care providers from the vaccine registry for the interview were the study population.

**Exclusion criteria**: Respondents absent from work due to different reasons were excluded during the interview.

## Sample size and sampling procedures

The sample size was calculated using EPI info version 7.1 stat calc for population survey, considering the following assumptions; P = the proportion of individuals who developed side effect after taking Asterzenica in Saudi Arabia in the community assumed to be 34.7 [25], 4% of margin of error (d), 95% of Confidence Interval (CI) for the desired confidence level (z), design effect of 1.5 and adding 15% of non-response rate the final sample was 939.

The list of the participants who received at least one dose COVID-19 vaccine any doses were obtained from the health facilities. The sample size was proportionally allocated to the selected facilities per total numbers of individuals who received vaccines. The participants were approached using systematic random sampling. The phone number of the participants was obtained from the registry to communicate the individuals selected for the interview before a day.

## Study variables

**Dependent variables**: COVID-19 vaccine side effect (in the $1^{st}$ dose or $2^{nd}$ dose).

**Independent variables**: Socio-demographic variables, comorbidities, behavioral factors like smoking history, Alcohol consumptions, dietary habit and physical activity, knowledge toward COVID_19 vaccines and other variables.

## Data collection procedures

The questionnaires were developed based on reviewing different literatures and guidelines used to assess the adverse effects of vaccines, intention to use the vaccines, previously used and validated for study [23–30,32–44]. The questionnaires were prepared in English and translated into Afan Oromo (local language). Twenty one data collectors with health professionals with a minimum of bachelor degree were participated for the field work and six master holder health professionals were recruited as supervisor. One day was given to the data collectors and supervisors on how to fill out the questionnaire and interview the respondents. A pretest was done on 48 individuals (5% of the total samples) selected from Asella Health Center (13 subjects), Eteya Health Center (1 0 subjects), Sagure Health Center (10 and Adama Health Center (15 subjects) and an amendment was made on the structure of the questions. Strict supervision was done by supervisors during fieldwork. The study participants and data collectors wore masks and kept acceptable physical distancing during data collection.

## Data entry and analysis

The data were checked for completeness and consistency then entered into Epi Info version 7.1 and exported to SPSS version 21 for analysis. Descriptive statistics were performed to describe the study population. The normality of the data was checked using a Q-Q plot. Logistic regression analysis was run to identify associated factors. Multi-collinearity was assessed

using variance inflation factor (VIF), and variables with VIF> 10 were removed from the analysis. The model fitness for logistic regression was tested using the Hosmer-Lemeshow goodness of fit test at P-Value >0.2. Bivariate analysis (one independent with dependent variable) was undertaken in binary logistic regression to determine the crude odds ratio of all risk factors independently and variables with p-value<0.2 were selected and entered into multiple logistic regressions to control confounders and to know the independent predictor of hypertension. Finally, the association was expressed in an adjusted odds ratio (AOR) with a 95% confidence interval, and P-value <0.05 was used as the cut-off point to declare significance in the final model.

## Ethical considerations

The study protocol was approved by the Institutional Review Board of Arsi University and Oromia Regional Health Bureau. The purposes and objectives of the study were explained in detail to all the participants and verbal consent was taken. The identity of the respondents was kept confidential to ensure privacy and for encouraging accurate responses to the questions.

## Result

### Socio-demographic characteristics

A total of 912 health workers participated in the study, with a response rate of 97.1%. The participant in each gender is almost similar [462(50.7% male, 450(49.3%) female]. The mean (±standard deviation) age of healthcare workers was 37.7(±7.2) years, and the majority of the respondents found within the age range of 30–39 years. The majority of them (63.7%) were married. Three hundred twenty-eight (36.07%) participants were Nurses by profession and 728(79.3%) were degree holders. With regard to religion, 474(52.0%) were Orthodox Christian religion followers followed by Muslims (216(23.7%) (S1 Table).

### Chronic comorbidities conditions of participants

Of the total participants, 57(6.3%) self-reported that they had chronic comorbidity. Of those who had chronic health problems, 26(45.6%), 16(28.1%), and 13(22.8%) had diabetes, hypertension and Asthma, respectively. Nearly three-quarter (73.7%) of the respondents were taking treatments for the comorbidities (S2 Table).

### COVID-19 status of the participants before the vaccination

Concerning COVID-19 test before vaccination, 398(43.6%) were tested and 136(34.2%) were positive. The majority (60.0%) the respondents received home-based treatment and 38(27.9%) at the COVID-19 treatment center. About one-fifth (19.4%) of the family members of the respondents ever infected with COVID-19 and 20(2.2%) of the respondents had family (relative) loss because of COVID-19 (S3 Table).

### Behavioral factors

About 25(2.7%) of the participants were smokers. and 183 (20.1%) had ever drunk some kind of alcohol; of these (36.4%) drank 1 to 3 days per month. About 80 (8.8%) had never chewed khat and 27.5% of them chew daily and less than once a month only 1.8% of the respondents took any drug other than cigarettes, alcohol, or khat, which was shisha. For 845 (92.7%) of the respondents their work did not involve vigorous-intensity activities that increase respiration for at least ten minutes. Only 67 (7.3%) of the respondent did vigorous-intensity physical activity, of these, 27(40.3%), 19 (28.4.3%), 13 (19.4%) did 2 days, 3days, and 4 days per week,

respectively. Meanwhile, about 124 (13.6%) of the respondents involve moderate physical activity, of whom 63(50.8%) spent five days per week (S4 Table).

## Preventive practice to protect COVID-19 infections pre and post COVID-19 vaccination

Participants were asked about their ever practice to protect against COVID-19 infection before and after receiving the COVID-19 vaccine during the pandemic. About 625(68.5%) and 564 (61.8%) avoided handshaking, hugging, and kissing in order to prevent contracting and spreading COVID-19, before and after receiving the vaccine, respectively. For the question were/are you frequently wash your hands with soap in order to prevent contracting and spreading COVID-19, 796(87.3%) and 778(85.3%) did it before and after the COVID-19 vaccine, respectively. With regard to their practice of using sanitizer/Alcohol in order to prevent contracting and spreading COVID-19, 784(14.0%) use sanitizer but after the vaccine the habit slightly decreased, 76.3% of the health professionals use sanitizer. All of the respondents wore masks before receiving the vaccine though the frequency varies, after receiving the vaccine about 32(3.5%) of the professionals did not wear mask totally. Over half (53.1%) of the respondents did not go to crowded places before receiving the COVID-19 vaccine moreover, the majority(92.8%) of the respondent go to crowded places after receiving COVID-19 Vaccine. Herbal products and traditional medicines were used by three quarters (75.7% of the respondents before vaccination and 21.7% after vaccination (Table 1).

## Participants knowledge about COVID-19 vaccine

Half (50.3%) of the respondents suspect the effectiveness of COVID-19 vaccination in the prevention and control of COVID-19 transmissions. About 746(81.8%) of the respondents said that anybody is eligible for COVID-19 vaccination and 102(11.2%) answered health workers are eligible for vaccination. All the study participants said that the COVID-19 vaccines have side effects and of the most mentioned side were nausea (95.6%), fatigue (95.4%), headache (91.6%), blood clotting problem (82.8%), joint pain (90.6%), and back pain (88.9%)(Table 2).

## Post -COVID-19 vaccine complications

As displayed in Fig 1 below the most reasons to receive COVID-19 vaccination were fear of being infected (89.5%) followed by not to spread infections to others (38.9%) and believe that the vaccine is very safe (29.3). Asterzenica vaccine was given in the 1st dose for the majority of the respondents. Over two- third (68.9) of the respondents took the 2nd dose was 68.9%. Among the respondent who received 2nd dose, 79.6% received after two months, (17.8%) after one month and 16(2.5%) did not know the time interval.

   Among those respondents who received only one dose, 157(55.3%) were willing to take the 2nd dose and 127(44.7) did not show a willingness to take the second dose. The vaccine produced in the USA was preferred by 91(58.0%) of the respondents to take the 2nd dose followed by the vaccine produced in Europe, preferred by 53(33.8%) of the participants. The most reasons mentioned by the respondents who did not intend to take the 2nd dose were fear of the side effects (96.1%), the concern about the rigor of testing of the vaccine(89.80%), the vaccine being ineffective(75.6%), the vaccine itself cause COVID-19(12.6%) (Fig 2).

   A total of 765 (84.0%) and 449(71.5%) participants experienced at least one side effects in the 1st and 2nd dose of the vaccines, respectively. The most reported side effects were pain at vaccination site(96.5% 1st dose Vs 65.6% 2nd dose), joint pain (88.3% 1st dose Vs 37.7% 2nd dose), fatigue (86.2% 1st dose Vs54.6% 2nd dose), back pain (83.2% 1st dose Vs 33.0% 2nd dose), headache 78.5% 1st dose vs 27.1% 2nd dose), chills (64.6%1st dose Vs 10.6% 2nd dose,

**Table 1. Respondents practice to protect COVID-19 infections pre and post COVID-19 vaccination for the study of post COVID-19 vaccine evaluations in Oromia region, Ethiopia, 2021.**

| Characteristics | Pre-COVID-19 vaccination (N %) | Post COVID-19 vaccination (N %) |
|---|---|---|
| Avoided hand shaking, hugging and kissing in order to prevent contracting and spreading COVID-19. | | |
| Yes | 625(68.5) | 441(48.4) |
| No | 287(31.5) | 471(51.6) |
| Frequently wash your hands with soap in order to prevent contracting and spreading COVID-19 | | |
| Yes | 796(87.3) | 548(60.1) |
| No | 116(12.7) | 364(39.9) |
| Are using sanitizer/Alcohol in order to prevent contracting and spreading COVID-19? | | |
| Yes | 784(86.0) | 564(61.8) |
| No | 128(14.0) | 348(38.2) |
| Use facial masks | | |
| Never | 0 | 32(3.5) |
| Only in public and crowded places | 199(21.8) | 169(18.5) |
| Most of the time | 259(28.4) | 334(36.6) |
| Always | 325(35.6) | 247(27.1) |
| When leaving home | 129(14.1) | 130(14.3) |
| Gone to any crowded place | | |
| Yes | 428(46.9) | 846(92.8) |
| No | 484(53.1) | 123(13.5) |
| Met up with friends or family (within 2 meter distance) | | |
| Yes | 598(65.6) | 853(93.5) |
| No | 314(34.4) | 59(6.5) |
| Use herbal products and traditional medicine, | | |
| Yes | 690(75.7) | 198(21.7) |
| No | 222(24.3) | 714(78.3) |

sleeping disorder(61.9% 1st dose Vs 24.7%), fever(52.5% 1st dose vs 28.4% 2nd dose),night mare (52.1% 1st dose Vs 13.9% 2nd dose), nausea(45.2% 1st dose Vs 12.6 2nd dose) and redness at vaccination site(29.1% 1st dose vs 13.4% 2nd dose).

Eight hundred forty (92.1%) of the participants experienced at least one side effect either in the first dose or second dose. On the other hand, three hundred seventy-five (59.0%) of the respondents experienced side effects in both doses.

Of the participant who experienced side effects, 226(29.5%) had treatment in the 1st dose, and 52(11.5%) in the 2nd dose. Only 1.8% and 0.7% of the respondents were hospitalized due to the side effects in the 1st dose and 2nd dose, respectively.

Of those who developed side effects, 367 (47.9%) and 235(51.8%) side effects emerged within the 1–4 hours in the first dose and 2nd dose of the COVID-19 vaccine. In the first dose, 50.1% of the side effects lasted for 1 to 5 days similarly, the majority (91.0%) of the side effects resolved within 1 to 5 days in the second dose of the vaccine (Table 3).

## Perception toward COVID-19 vaccination and Re-infection of COVID-19

Two hundred five (22.5%) respondents suspect the effectiveness and 74.3% of the respondents were worried about the negative effects of the vaccine they vaccinated. About one-fourth 226

**Table 2. Participants knowledge about COVID-19 Vaccine for the study of post COVID-19 vaccine evaluations in Oromia region, Ethiopia, 2021.**

| Characteristics | Frequency | Percentage |
|---|---|---|
| **Is vaccination an effective way to prevent and control COVID-19 transmissions** | | |
| Yes | 383 | 42.0 |
| No | 70 | 7.7 |
| Probable | 459 | 50.3 |
| **Who are eligible for COVID-19 vaccination** | | |
| Health workers | 102 | 11.2 |
| Persons with chronic illnesses | 20 | 2.2 |
| Elderly | 44 | 4.8 |
| Any body | 746 | 81.8 |
| **How many doses of COVID-19 vaccine are given** | | |
| One dose | 167 | 18.3 |
| Two does | 708 | 77.6 |
| I do not Know | 37 | 4.1 |
| **Do you think COVID-19 Vaccine has side effect** | | |
| Yes | 912 | 100 |
| No | 0 | 0 |
| **Side effect** | | |
| Blood clotting problem | 755 | 82.8 |
| Pain at vaccination sites | 610 | 66.9 |
| Redness at the vaccination site | 236 | 25.9 |
| Fever | 473 | 51.9 |
| Fatigue | 870 | 95.4 |
| Abdominal pain | 322 | 35.3 |
| Diarrhea | 263 | 28.8 |
| Vomiting | 389 | 42.7 |
| Nausea | 872 | 95.6 |
| Difficulty of swallowing | 297 | 32.6 |
| Cough | 123 | 13.5 |
| Chills | 516 | 56.6 |
| Night mare | 619 | 67.9 |
| Headache | 835 | 91.6 |
| Sleeping disorder | 525 | 57.6 |
| Back pain | 811 | 88.9 |
| Joint Pain | 826 | 90.6 |

(24.8%) of the participants disagree that the vaccine will not end up the pandemic. The majority of the respondents felt some fear when they were vaccinated COVID-19 vaccine and 290 (31.8%) were afraid of infection and getting sick after vaccination, 281(30.8%) felt a little afraid of becoming infected after infection and 341(37.4) did not fear infection after vaccination. One hundred forty-two (15.6%) respondents were tested for COVID-19 after vaccination and 21(14.8%) were positive for the test meanwhile 1(4.8%) were admitted in hospital or treatment center (Table 4).

## Factors associated with Post COVID-19 Vaccination reported side effects

Ever drink alcoholics, religion, engaging in moderate-intensity physical activity, felt fear when vaccinated were variables which showed significant association with post-COVID-19

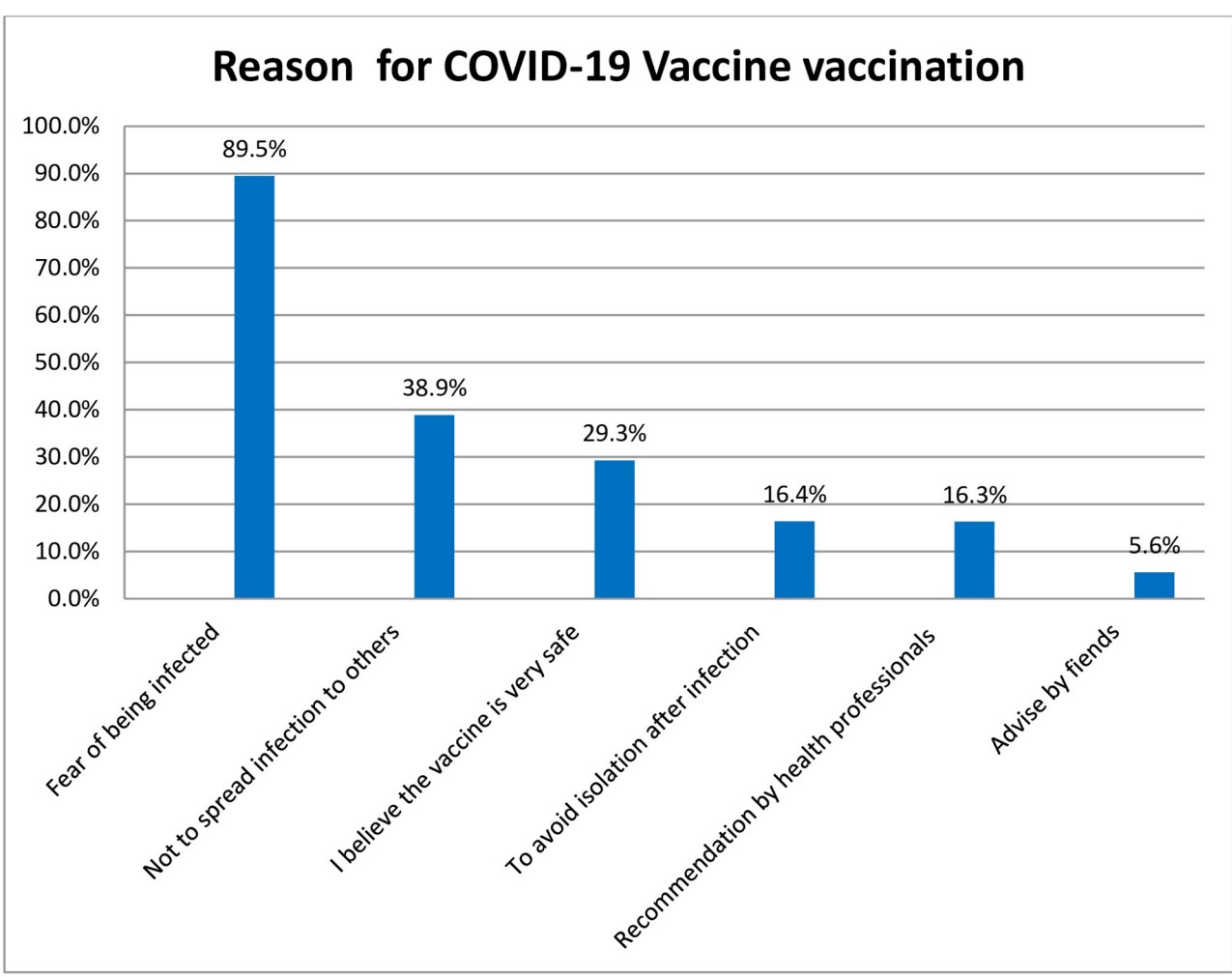

**Fig 1. Reasons of COVID-19 Vaccine vaccination among participants for the study of COVID-19 vaccine complication evaluations in central Oromia region, Ethiopia, 2021.**

vaccination side effects in the bivariate analysis. Variables with P-value <0.2 were included in the multivariate analysis to control confounding, but only engaging in moderate-intensity physical activity and feeling fear when vaccinated were variables that remained significantly associated with post-vaccination side effects.

Respondents who did not engage in physical activity were 7.54 fold more likely to develop post-COVID-19 vaccination side effects compared to those who involved at least moderate-intensity physical activity [AOR = 7.54, 95% CI;2.46,23.12]. The odds of experiencing side effects among the respondents who felt fear when vaccinated were 10.73 times compared not felt fear (AOR = 10.73, 95% CI; 2.47,46.64), and similarly, those who felt little fear were 4.28 times more likely to develop side effects (AOR = 4.28, 95% CI; 1.28, 14.39) (Table 5).

## Discussions

This study tried to show self-reported post-COVID-19 vaccine adverse effects among health care workers in the central Oromia region in 2021. The study tried to compare the events before and after the vaccine preventive measures toward COVID-19 infection. Importantly

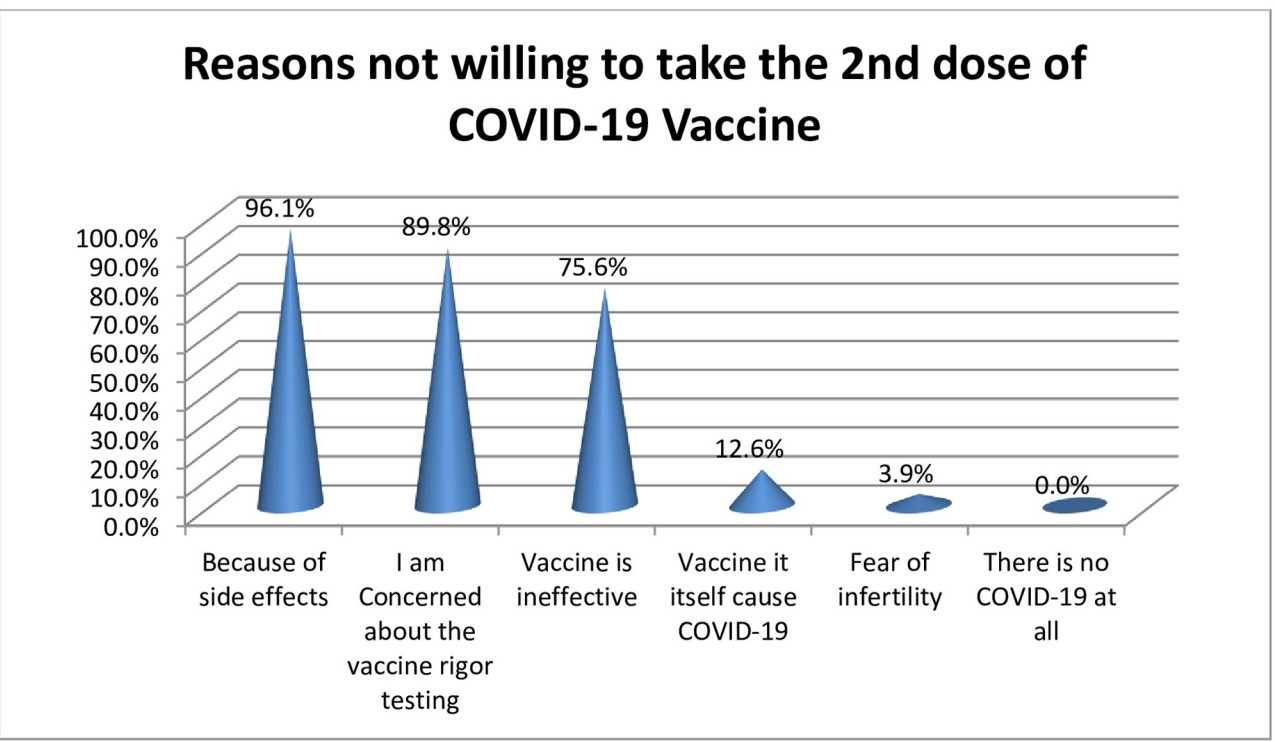

**Fig 2. Reasons not willing to take the 2nd dose of COVID-19 vaccine among health professionals for the study of post COVID-19 vaccine complication evaluations in central Oromia region, Ethiopia, 2021.**

the study tried to identify the adverse effects post 1st dose and 2nd dose of the COVID-19 vaccines.

About four-ninth (44.7%) of the study participants did not show a willingness to take the 2nd dose of COVID-19 vaccines after the 1st dose. The finding is higher than the study in Togo, where 10.9% of the respondents were not ready to take the second dose of the vaccine [45]. This could be due to the fear of the side effects since a large proportion of the respondents experienced at least one side effect in the 1st dose. The other speculation might be due to the inadequacy of information provided by the vaccine providers (injectors) about the risks and benefits of taking the vaccine.

This study reported that 84.0% and 71.5% of participants experienced at least one side effect in the 1st and 2nd dose of the vaccines, respectively. In contrast to our findings, a study in Korea among BNT162b2 mRNA COVID-19 vaccine receivers showed that the rates of adverse reactions were higher after the 2nd dose compared with the 1st dose (89.1% vs. 80.1%) [33]. The difference could be due to the types the vaccine provided, in our cases majority of the participants received the AstraZeneca vaccine where the side effect is pronounced in the first jab compared to the subsequent dose.

In our study, 84.0% of the participants reported at least one side effect post-vaccination in the 1st dose. Similar findings were reported in the previous studies in Slovakia and German health workers; 85.8% of the participants receiving the BNT162b2 vaccine in Slovakia reported at least one side effect [28] and 88.1% of the healthcare workers in Germany reported at least one side effect after receiving COVID-19 vaccines; 87.1% following mRNA-based vaccines and 92% following the viral vector-based vaccine [2]. The studies conducted in Saudi Arabia, the USA, Israel, and Togo reported slightly lower side effects compared to our findings. The

**Table 3. Participants self-reported COVID-19 vaccine side effects for the study of post COVID-19 vaccine evaluations in Oromia region, Ethiopia, 2021.**

| Characteristics | 1st dose of COVID-19 (N %) | 2nd dose of COVID-19 (N %) |
|---|---|---|
| **Type of COVID-19 you received** | | |
| Astrazenica | 527(57.8) | 452(72.0) |
| Johnssons & Johnsons | 147(16.1) | 71(11.3) |
| Sinopharm | 14(1.5) | 26(4.1) |
| I do not know | 224(24.6) | 79(12.6) |
| **Side effect** | | |
| Yes | 765(84.0) | 449(71.5) |
| No | 146(16.0) | 179(28.5) |
| **Types of side effects** | | |
| Headache | 601(78.5) | 123(27.1) |
| Fatigue | 660(86.2) | 248(54.6) |
| Joint Pain | 676(88.3) | 171(37.7) |
| Night mare | 399(52.1) | 63(13.9) |
| Back pain | 637(83.2) | 150(33.0) |
| Sleeping disorder | 474(61.9) | 112(24.7) |
| Pain at vaccination sites | 739(96.5) | 163(65.6) |
| Redness at the vaccination site | 223(29.1) | 61(13.4) |
| Fever | 402(52.5) | 129(28.4) |
| Abdominal pain | 98(12.8) | 29(6.4) |
| Diarrhea | 78(10.2) | 25(5.5) |
| Vomiting | 44(5.7) | 24(5.3) |
| Nausea | 346(45.2) | 57(12.6) |
| Throat pain | 32(4.2) | 17(3.7) |
| Cough | 97(12.7) | 23(5.1) |
| Chills | 495(64.6) | 48(10.6) |
| Skin rash | 20(2.6) | 2(0.4) |
| **Time to happen side effects** | | |
| Within 1–4 hours | 367(47.9) | 235(51.8) |
| Within 5–8 hours | 210(27.4) | 122(26.9) |
| Within 9–12 hours | 131(17.1) | 53(11.7) |
| After 12 hours | 58(7.6) | 44(9.7) |
| **Duration of side effects** | | |
| 2-5days | 384(50.1) | 413(91.0) |
| 6–10 days | 207(27.0) | 35(7.7) |
| 11–20 days | 125(16.3) | 6(1.3) |
| >20 days | 50(6.5) | - |
| **Received any treatment for the side effects** | | |
| Yes | 226(29.5) | 52(11.5) |
| No | 540(70.5) | 402(88.2) |
| **Hospitalized/ admitted to hospital due to the side effects** | | |
| Yes | 14(1.8) | 3(0.7) |
| No | 752(98.2) | 451(99.3) |

**Table 4. Perceptions related to COVID-19 vaccine and Re-infection among participants for the study of COVID-19 Vaccine complication evaluations in central Oromia region, Ethiopia, 2021.**

| Characteristics | Frequency | Percentage |
| --- | --- | --- |
| **Opinion regarding the effectiveness of the vaccine received** | | |
| Effective | 407 | 44.6 |
| Ineffective | 205 | 22.5 |
| I have no opinion | 300 | 32.9 |
| **Worried about the negative effects of taking the vaccine** | | |
| Yes | 678 | 74.3 |
| No | 234 | 25.7 |
| **Agree that taking the vaccine will help to prevent the pandemic** | | |
| Agree | 424 | 46.5 |
| Neutral | 258 | 28.3 |
| Disagree | 226 | 24.8 |
| Strongly disagree | 4 | 0.4 |
| **Felt fear, when vaccinated** | | |
| Yes | 348 | 38.2 |
| Little | 383 | 42.0 |
| No | 181 | 19.8 |
| **Afraid of infection and getting sick after vaccinated** | | |
| Yes | 290 | 31.8 |
| Little | 281 | 30.8 |
| No | 341 | 37.4 |
| **Have you tested for COVID-19 after vaccine** | | |
| Yes | 142 | 15.6 |
| No | 770 | 84.4 |
| **If yes (tested) what was the result** | | |
| Negative | 121 | 85.2 |
| Positive | 21 | 14.8 |
| **Admitted in hospital or treatment center** | | |
| Yes | 1 | 4.8 |
| No | 20 | 95.2 |

studies conducted in Saudi Arabia, 60%, 68%, and 34.7% of the study subjects after receiving Oxford-AstraZeneca and Pfizer-BioNTech vaccines, reported at least one side effect [23–25]. In a study in the USA, 64.5% received the BNT162b2 mRNA vaccine, Israel 77% (44% of males and 84.5% of females), and in Togo, 71.6% of health care workers reported at least one adverse effect [30,33,45]. The difference could be due to the study setting, methods of interview, immune response variation and pre-awareness used by providers before the administration of the vaccine used might be different from Ethiopia healthcare workers. When compared to the study conducted in Vietnam, the reported side effect is slightly lower, 96.1% of the respondents in Vietnam reported at least one side effect [29].

Injection site pain was the most prevalent local side effect, 96.5% in the 1st dose and 65.6% 2nd dose respondents reported this side effect. In support of our finding, different studies reported comparable figures where a significant percentage of the respondents who received the COVID- 19 vaccine suffered from non-life-threatening injection site pain. In a study in India, the majority experienced pain at the injection site (88.8–100%), post-vaccination of COVID- 19(35). A Slovakian participant reported 85.2% of injection site pain [28]. In studies in Germany, injection site pain was the most prevalent local side effect (75.6% and 70.2%)

**Table 5. Associated factors with post COVID-19 vaccination side effects among the participants for the study of post COVID-19 vaccination evaluation in central Oromia region Ethiopia, 2021.**

| Characteristics | Experienced at least one side effects | | Odds ratio with 95% CI | | |
|---|---|---|---|---|---|
| | No (N %) | Yes (N %) | COR | P-value | AOR |
| **Sex** | | | | 0.696 | |
| Male | 34(7.4) | 425(92.6) | 1 | | - - |
| Female | 30(6.7) | 415(93.3) | 1.11(0.67,1.84) | | |
| **Age** | | | | 0.168 | |
| 20–29 | 18(5.6) | 306(94.4) | 0.99(0.42.2.32) | | - - |
| 30–39 | 38(8.8) | 396(91.2) | 0.60(0.30,1.33) | | - - |
| > = 40 | 8(5.5) | 138(94.5) | 1 | | |
| **Profession** | | | | 0.198 | |
| Physician(MD) | 6(5.5) | 104(94.5) | 1 | | - |
| Pharmacist | 15(10.9) | 122(89.1) | 0.95(0.28,3.23) | | |
| Health officer | 5(12.2) | 36(87.8) | 0.45,0.16,1.27) | | |
| Nurse | 16(5.0) | 306(95.0) | 0.40(0.11,1.45) | | |
| Midwifery | 15(9.1) | 150(90.9) | 1.05(0.40,2.95) | | |
| Anesthetist | 2(5.2) | 31(93.9) | 0.55(0.19,1.56) | | |
| Laboratory | 5(5.2) | 91(94.8) | 0.85(1.6,4.61) | | |
| **Religion** | | | | 0.003 | |
| Orthodox | 20(4.3) | 450(95.7) | 2.83(1.53,5.24) | 0.308 | 2.30(0.62,8.60) |
| Other [z] | 20(9.1) | 199(90.9) | 1.25(0.67,2.34) | | 0.89(0.25,3.15) |
| Muslim | (11.2) | 191(88.8) | 1 | | 1 |
| **Presence of comorbidities** | | | | 0.422 | |
| No | 62(7.3) | 785 (92.7) | 1 | | |
| Yes | 2(3.5) | 55(96.5) | 2.17(0.52,9.12) | | |
| **COVID-19 test result** | | | | | |
| Negative | 15(5.7) | 247(94.3) | 1 | 0.065 | |
| Positive | 2(1.5) | 134(98.5) | 4.10(0.92,18.10) | | |
| **Drink alcoholic** | | | | 0.023 | |
| No | 58(8.1) | 661(91.9) | 1 | | 1 |
| Yes | 6(3.2) | 179(96.8) | 2.62(1.11,6.20) | | 2.32(0.46,11.73) |
| **Chew Khat** | | | | | |
| No | 62(7.5) | 762(92.5) | 1 | 0.094 | |
| Yes | 2(2.5) | 78(97.5) | 3.17(0.76,13.22) | | - - |
| **Involve moderate-intensity activity** | | | | 0.0001 | |
| No | 45(5.8) | 736(94.2) | 3.00(1.70,5.31) | | *7.54(2.46,23.12)* |
| Yes | 19(15.4) | 104(84.6) | 1 | | 1 |
| Feel fear when vaccinated | | | | 0.0001 | |
| Yes | 12(3.5) | 328(96.3) | 5.87(2.94,11.72) | | *10.73(2.47,46.64)* |
| Little | 20(5.2) | 363(94.8) | 3.90(2.16,7.03) | | *4.28(1.28,14.39)* |
| No | 32(17.7) | 149(82.3) | 1 | | 1 |

Other; Protestant, catholic, waqefeta.

[2,35]. In a study conducted in Saudi, 85% of the participants felt pain at the site of the injections [23] and other studies conducted in Saudi reported that 41.2% and 30.5% injection site pain prevalence [24,25]. A stud in Vietnam showed that 58.3% of the participants reported pain at the injection site [29]. A study was conducted in Israel. Injection site pain was the most common symptom [33]. Meanwhile, 29.1% in the 1st dose and 13.4% in the 2nd dose were felt

redness at the vaccination site which is higher than the finding in Indonesia only 2.0% and showed redness after injection of the vaccine [36] and in Germany, about 18% developed injection swelling and (10.4%) injection site redness [2]. In a study in the United Arab Emirates on the effects of the evidence on Sinopharm COVID-19 vaccine side effects; pain at the vaccination site, fatigue, headache, and tenderness were the most common side effects post-second dose [37].

Joint pain was reported by 88.3% of respondents in the 1st dose and by 37.7% in the 2nd dose. In addition to this 83.2% of participants in the 1st dose and 33.0% in the 2nd dose complained of back pain. The finding is supported by the study conducted in Jordan [38]. The finding is much higher than the studies conducted in Saudi Arabia, 23% and 27.5% reported a symptom of joint pain [24,25] as well in Indonesia study where only 1.5% of the respondents reported joint pain post-vaccination [36].

In our study, fatigue was reported by 86.2% of respondents in the 1st t dose and 54.6% in the 2nd dose. In concomitant to this finding two studies conducted in Saudi Arabia, 90% and 36.1% of participants reported fatigue [23,24], in Israel, fatigue was reported by 65.7% of the participants [32]. In a study in India, (87.7–60%) experienced tiredness and 86.6% body ache [34]. In a study in Germany after vaccination with the ChAdOx1 nCoV-19 (AZD1222) vaccine, 44.8% of the reported side effect was fatigue [38] and another study in German reported fatigue of 34.7% among the participants [35]. In a study in Slovakia among receiving the BNT162b2 vaccine, fatigue was reported by 54.2% of the receiver [28]. In a study in South Korea, the commonest systemic reactions to the ChAdOx1-S and the BNT162B2 vaccines are muscle aches (77.7%) and fatigue (74.7%) [39]. A study in Vietnam indicates that 62.5% of the respondents reported fatigue [29]. Among participants from Togo, about three-quarters (74.3%) reported asthenia post-vaccine [45]. Fatigue was the most reported complaint after vaccine among Jordanian health care providers [38].

Over three quarters (78.5%) of the respondents in the 1st dose and 27.1% 2nd dose complained of headaches. The finding is supported by the study conducted in Europe, a study in Germany and Jordan reported that headaches were the most reported side effect after vaccination with the ChAdOx1 nCoV-19 (AZD1222) [27,35]. The finding is higher when compared to the study done in Indonesia, Saudi, Slovakia, Vietnam, Israel, and Germany, where 22.1%, 24.2%, 34.3%, 58.5%, 48.7%, and 30.6% of the respondents were suffered from headache [24,28,29,32,35,36]. A study in South Korea and Togo supports this finding where a headache was felt by 67.4% and 68.7% of the respondents, respectively [39,45]. A study from Jordan also supports this finding [38]. In previous study the most frequently reported neurological adverse event of ChAdOx1 nCoV-19 (AZD1222) vaccine [40]. The finding is comparable with 10 placebo recipient and one recipient of correct vaccine reported undesirable side effects such as injection site pain, fatigue and headache [41].

In this study, nearly two-thirds (64.6%) and 10.6% of the participants encountered chills in the 1st dose and 2nd dose, respectively, similar to this result of 63.5%, 36.1%, 26.4%,45.7% and 44.2% of the participants from South Korea, Germany, Slovakia, Vietnam, and Israel felt chills post-vaccination, respectively (26,28,29,33,39] and in addition, Jordanian respondents were felt chills post-vaccination [27].

Sleeping disorder is one of the manifestations reported by the study participants, 61.9% of the participants in the 1st dose and 24.7% in the 2nd dose companioned sleeping disorder in this study. The finding is complemented by the study in Vietnam, 62.5% of the subjects had the problem of sleeping disorder post-vaccination [29].

The most mentioned reasons not to take the 2nd dose in this study were; because of side effects (96.1%), the concern about the rigor of testing of the vaccines (89.80%), the vaccine is ineffective(75.6%), Vaccine itself cause COVID-19(12.6%). Similar to this study a study in the

United Arab Emirates; the most common reason for being unwilling to take the COVID-19 vaccine was that vaccines are not effective [37]. In contrast to this study, 97.61% of the USA participants intended to have the second dose and 92.9% had already received it [30]. The difference could be due to gap of the level of awareness between the USA participants and our setting.

Of those who developed side effects, (47.9%) and (51.8%) of side effects emerged within the 1–4 hours in the first dose and 2nd dose of COVID-19 vaccine. In the first dose, 50.1% of the side effects lasted for 2 to 5 days similarly majority (91.0%) of the side effects resolved within 2 to 5 days in post second dose of vaccine. The finding in line with the study in Slovakia among receiving the BNT162b2 vaccine, the reported side effects were of a mild nature (99.6%) and were resolved within three days [28].

About 30% of the participant received treatment for the side effects that occurred in the 1st dose and 11.5% in the 2nd dose. Similar to this study in the USA, (2.49%) required help from an outpatient provider, (0.62%) required help from an emergency department [30]. In the other way, in a study in Slovakia the reported side effects were of a mild nature (99.6%) that did not require medical attention, and a short duration, as most of them (90.4%) were resolved within three days [28]. The rate of hospitalization was 1.8% in the 1st dose and 0.7% in the 2nd dose which is similar to the study in Saudi; the rate of hospitalization was 1% [38] and in USA 0.25% patients required hospitalization [30]. In a study in Korea, 3.3% visited the emergency department to seek treatment for the side effects after vaccination, but none of the respondents were hospitalized [42]. A study in Togo, of the participants who experienced adverse events, 18.2% were unable to go to work the day after vaccination, 10.5% consulted a medical doctor, and 1.0% were hospitalized [45].

About three-quarters (74.3%) of the respondents were worried about the negative effects of the COVID-19 vaccine they vaccinated. Similar to this finding in a study Togo, 67.5% of respondents expressed concern about long-term adverse events [45]. Majority of the respondent felt some fear when they were vaccinated COVID-19 vaccine.

Concerning the feeling of the respondents while taking vaccines, the majority of our respondents felt some form of fear, 31.8% were afraid of infection and getting sick after vaccination, 30.8% felt a little afraid of becoming infected after infection and 37.4% did not fear infection after vaccination. This result slightly disagrees with the study conducted among Polish doctors where few participants experienced a sense of fear, and almost 58% of the respondents did not feel it at all [43]. It is obvious that whether healthcare professionals or other individuals seen at clinic by health care providers feel some form of fear. The above behaviors may indicate not only a natural approach to the injection procedure, but also a human need to strengthen the condition of doctors to continue functioning in their daily work for the benefit of patients and personal safety.

Post COVID-19 vaccination infection was 14.8% among the respondents and 4.8% were admitted at hospital or treatment center due to the infection. Similar to this finding, 5.1% of the respondents were infected in the Aseer region, Kingdom of Saudi Arabia after vaccination [24]. A study in north India to see the occurrence of COVID-19 in priority groups receiving ChAdOx1 nCoV-19 coronavirus vaccine; 41% SARS-CoV-2 infection was observed after a single dose and (19%) re-infected after receiving both doses [44].

In the multivariate analysis respondents who did not engage in physical activity were 7.54 fold more likely to develop post-COVID-19 vaccination side effects compared to those who involved at least moderate-intensity physical activity. It is well known that active physical activity reduces stress, boosts the immune system, and prevents different forms of comorbidities; especially chronic non-communicable diseases [46]. The odds of experiencing side effects among the respondents who felt fear when vaccinated were 10.73 times compared not

felt fear), and similarly, those who felt little fear were 4.28 times more likely to develop side effects. It is advisable to provide a brief explanation of the vaccine-related side effects before providing the vaccine, to reduce side effects related to obsession.

Despite this study reporting interesting findings, it has some limitations. One of the limitations of this study is the problem of the study design. In a survey study, it is difficult to assess the outcomes directly; there might be a problem with temporal relationships and other draws back of the design. The other limitation subjection is subject to some forms of bias, like recall bias and social desirability bias. The severity and intensity of adverse reactions were not graded, i.e., there may have been overestimated or underestimated due to the nature of the self-reporting survey. The reported adverse reactions in this study were not medically attended adverse events immediately after injection, and not documented at facilities properly. This study only considered health workers, other population groups was underrepresented and the generalizability of the findings to other population groups is limited.

## Conclusions and recommendations

Most of the side effects reported by the respondents were not life-threatening. The majority (92.1%) of the participants experienced at least one side effect either in the 1st dose or 2nd dose, (84.0%) and (71.5%) participants experienced at least one side effect in the 1st and 2nd dose of the vaccines, respectively. Engaging in moderate physical activity and feeling fear when vaccinated were the characteristics independently associated with reported side effects post-vaccination.

Since the study is conducted retrospectively, it is difficult to characterize the side effects. It is better to plan monitoring and evaluation of the post-vaccine effect using standard longitudinal study designs to measure the side effects directly among different groups of the population to see the effectiveness of the COVID-19 vaccines administered to the population in preventing the transmission of SARS-COV2 infection. It is highly recommended to provide health education for the clients about the benefits and the side effects of the vaccines before administering the vaccine to reduce observed anxiety related to the vaccine that could occur after taking the vaccine since the vaccine is under evaluation. In addition, further research on the cold chain of the vaccine, the reported side effect may be due to the problem related to the storage, transportation, and problem related to the administration of the vaccine thought this factor not addressed in this study.

## Supporting information

**S1 Table. Socio-demographic characteristics of respondents with comorbidities for the study of post COVID-19 vaccine evaluations in Oromia region, Ethiopia, 2021.**
(DOCX)

**S2 Table. Chronic health problems of the respondents for the study of post COVID-19 vaccine evaluations in Oromia region, Ethiopia, 2021.**
(DOCX)

**S3 Table. Respondents COVID-19 infection status before the vaccination for the study of post COVID-19 vaccine evaluations in Oromia region, Ethiopia, 2021.**
(DOCX)

**S4 Table. Respondents behavioral factors for the study of post COVID-19 vaccine evaluations in Oromia region, Ethiopia, 2021.**
(DOCX)

**S1 Dataset.**
(SAV)

## Acknowledgments

We would like to express our deepest gratitude to the study participants, data collectors and supervisors for providing important information. We would also like to thank hospital administrators who helped us during the data collection period in facilitations.

## Author Contributions

**Conceptualization:** Mesfin Tafa Segni, Hailu Fekadu Demissie, Muhammedawel Kaso Adem, Adem Kedir Geleto, Birhanu Kenate Sori, Melese Lemmi Heyi, Dhabesa Gobena Iticha, Gemechu Shumi Bejiga, Abera Botere Guddisa, Yadeta Ayana Sima.

**Data curation:** Mesfin Tafa Segni, Hailu Fekadu Demissie, Muhammedawel Kaso Adem, Adem Kedir Geleto, Birhanu Kenate Sori, Melese Lemmi Heyi, Dhabesa Gobena Iticha, Gemechu Shumi Bejiga, Abera Botere Guddisa.

**Formal analysis:** Mesfin Tafa Segni, Hailu Fekadu Demissie, Muhammedawel Kaso Adem, Adem Kedir Geleto, Birhanu Kenate Sori, Melese Lemmi Heyi, Dhabesa Gobena Iticha, Yadeta Ayana Sima.

**Funding acquisition:** Mesfin Tafa Segni, Hailu Fekadu Demissie, Muhammedawel Kaso Adem, Adem Kedir Geleto, Birhanu Kenate Sori, Melese Lemmi Heyi, Dhabesa Gobena Iticha, Gemechu Shumi Bejiga, Abera Botere Guddisa, Lemessa Tadesse Amente, Mengistu Bekele Hurisa.

**Investigation:** Mesfin Tafa Segni, Hailu Fekadu Demissie, Muhammedawel Kaso Adem, Adem Kedir Geleto, Mesfin Wubishet Kelkile, Melese Lemmi Heyi, Dhabesa Gobena Iticha, Gemechu Shumi Bejiga, Yadeta Ayana Sima, Lemessa Tadesse Amente, Tesfaye Kebebew Jiru.

**Methodology:** Mesfin Tafa Segni, Hailu Fekadu Demissie, Muhammedawel Kaso Adem, Adem Kedir Geleto, Mesfin Wubishet Kelkile, Dhabesa Gobena Iticha.

**Project administration:** Mesfin Tafa Segni, Hailu Fekadu Demissie, Muhammedawel Kaso Adem, Adem Kedir Geleto, Mesfin Wubishet Kelkile, Birhanu Kenate Sori, Melese Lemmi Heyi, Gemechu Shumi Bejiga, Abera Botere Guddisa, Lemessa Tadesse Amente, Dereje Abdena Bayisa, Mengistu Bekele Hurisa.

**Resources:** Mesfin Tafa Segni, Hailu Fekadu Demissie, Adem Kedir Geleto, Birhanu Kenate Sori, Melese Lemmi Heyi, Dhabesa Gobena Iticha, Gemechu Shumi Bejiga, Abera Botere Guddisa, Lemessa Tadesse Amente, Dereje Abdena Bayisa, Mengistu Bekele Hurisa, Tesfaye Kebebew Jiru.

**Software:** Mesfin Tafa Segni, Adem Kedir Geleto, Birhanu Kenate Sori, Yadeta Ayana Sima, Dereje Abdena Bayisa.

**Supervision:** Mesfin Tafa Segni, Hailu Fekadu Demissie, Mesfin Wubishet Kelkile, Birhanu Kenate Sori, Melese Lemmi Heyi, Dhabesa Gobena Iticha, Abera Botere Guddisa, Yadeta Ayana Sima, Lemessa Tadesse Amente, Dereje Abdena Bayisa, Mengistu Bekele Hurisa, Tesfaye Kebebew Jiru.

**Validation:** Mesfin Tafa Segni, Mesfin Wubishet Kelkile, Birhanu Kenate Sori, Melese Lemmi Heyi, Dhabesa Gobena Iticha, Gemechu Shumi Bejiga, Abera Botere Guddisa, Yadeta

Ayana Sima, Lemessa Tadesse Amente, Dereje Abdena Bayisa, Mengistu Bekele Hurisa, Tesfaye Kebebew Jiru.

**Visualization:** Mesfin Tafa Segni, Muhammedawel Kaso Adem, Mesfin Wubishet Kelkile, Birhanu Kenate Sori, Melese Lemmi Heyi, Gemechu Shumi Bejiga, Abera Botere Guddisa, Yadeta Ayana Sima, Lemessa Tadesse Amente, Dereje Abdena Bayisa, Mengistu Bekele Hurisa, Tesfaye Kebebew Jiru.

**Writing – original draft:** Mesfin Tafa Segni, Hailu Fekadu Demissie, Muhammedawel Kaso Adem, Mesfin Wubishet Kelkile, Birhanu Kenate Sori, Melese Lemmi Heyi, Yadeta Ayana Sima, Lemessa Tadesse Amente, Dereje Abdena Bayisa, Mengistu Bekele Hurisa, Tesfaye Kebebew Jiru.

**Writing – review & editing:** Mesfin Tafa Segni, Hailu Fekadu Demissie, Mesfin Wubishet Kelkile, Birhanu Kenate Sori, Yadeta Ayana Sima, Lemessa Tadesse Amente, Dereje Abdena Bayisa, Mengistu Bekele Hurisa, Tesfaye Kebebew Jiru.

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
