## [Decision Letter · Decision Letter 0]

12 May 2022

PONE-D-22-03691Post COVID-19 vaccine vaccination side effects and associated factors among vaccinated health care providers in Oromia region, Ethiopia in 2021PLOS ONE

Dear Dr. Tafa,

Thank you for submitting your manuscript to PLOS ONE. After careful consideration, we feel that it has merit but does not fully meet PLOS ONE’s publication criteria as it currently stands. Therefore, we invite you to submit a revised version of the manuscript that addresses the points raised during the review process.

We look forward to receiving your revised manuscript.

Kind regards,

Grzegorz Woźniakowski, Full professor, PhD, ScD

Academic Editor

PLOS ONE

Journal Requirements:

Reviewers' comments:

Reviewer's Responses to Questions

**Comments to the Author**

1. Is the manuscript technically sound, and do the data support the conclusions?

Reviewer #1: Yes

2. Has the statistical analysis been performed appropriately and rigorously? 

Reviewer #1: Yes

3. Have the authors made all data underlying the findings in their manuscript fully available?

Reviewer #1: No

4. Is the manuscript presented in an intelligible fashion and written in standard English?

Reviewer #1: No

5. Review Comments to the Author

Reviewer #1: ABSTRACT

The background section of the abstract is so verbose. The authors should shorten it to remove some irrelevant details.

The authors should include the inclusion and exclusion criteria as well as the outcome measures in the method section of the abstract.

The authors should avoid being too verbose in the recommendation for the study in the abstract. Why must they join Conclusion and recommendation as a heading? The heading should be only Conclusion

INTRODUCTION

The authors should delete so many redundant sentences . However, they should beef up the justification for the study.

METHODS

Why did the authors entered data into Epi Info version 7.1 and exported to SPSS version 21 for analysis? Epi info too can do analysis.

RESULTS

There are so many tables to the extent that it decreases global easy readership.

DISCUSSION

The discussion section appears to be a mere repetition of results. The authors should redo the discussion to reflect the real discussion of manuscript.

What are the clinical implications for the study findings. The authors should discuss it.

6. PLOS authors have the option to publish the peer review history of their article (what does this mean?). If published, this will include your full peer review and any attached files.

Reviewer #1: No

---

## [Author Response · Author response to Decision Letter 0]

30 Jul 2022

Dear reviewer 

Thank you for your time to review our manuscript ‘‘Manuscript ID: PONE-D-22-03691; Post COVID-19 vaccine vaccination side effects and associated factors among vaccinated health care providers in Oromia region, Ethiopia in 2021’’. We tried to incorporate your concern in the revised manuscript and the response is appended below .

With regards 

Mesfin Tafa Segni (corresponding author) 

Response to Reviewer 

Reviewer #1: ABSTRACT

The background section of the abstract is so verbose. The authors should shorten it to remove some irrelevant details.

The authors should include the inclusion and exclusion criteria as well as the outcome measures in the method section of the abstract.

The authors should avoid being too verbose in the recommendation for the study in the abstract. Why must they join Conclusion and recommendation as a heading? The heading should be only Conclusion

Response: The comment is accepted and revision is done highlighted red

INTRODUCTION

The authors should delete so many redundant sentences. However, they should beef up the justification for the study.

Response: Comment accepted and revision done redundancy is reduced as possible in the revised document.

METHODS

Why did the authors entered data into Epi Info version 7.1 and exported to SPSS version 21 for analysis? Epi info too can do analysis.

Response: Yes it is possible to do analysis using EPI Info. But it is good to use other software for analysis and for cleaning the data. EPI info is more powerful for data entry to reduce errors that occur during entry. 

RESULTS

There are so many tables to the extent that it decreases global easy readership.

Response: the tables reduced and the rest considered as supplementary materials in the revised version.

DISCUSSION

The discussion section appears to be a mere repetition of results. The authors should redo the discussion to reflect the real discussion of manuscript.

What are the clinical implications for the study findings? The authors should discuss it.

Response: The comment accepted and the revision done extensively in the revised version of the document.

.

---

## [Decision Letter · Decision Letter 1]

6 Sep 2022

PONE-D-22-03691R1Post COVID-19 vaccine vaccination side effects and associated factors among vaccinated health care providers in Oromia region, Ethiopia in 2021PLOS ONE

Dear Dr. Tafa,

Thank you for submitting your manuscript to PLOS ONE. After careful consideration, we feel that it has merit but does not fully meet PLOS ONE’s publication criteria as it currently stands. Therefore, we invite you to submit a revised version of the manuscript that addresses the points raised during the review process.

We look forward to receiving your revised manuscript.

Kind regards,

Mohd Adnan, PhD

Academic Editor

PLOS ONE

Additional Editor Comments:

Manuscript does not fulfill the standards established for the journal to be considered for publication in its current form. I agree with the reviewer that manuscript still requires substantial revision and additional work to support the conclusion and improve the quality of the publication. Please see the detailed comments made by the reviewers.

Reviewers' comments:

Reviewer's Responses to Questions

**Comments to the Author**

1. If the authors have adequately addressed your comments raised in a previous round of review and you feel that this manuscript is now acceptable for publication, you may indicate that here to bypass the “Comments to the Author” section, enter your conflict of interest statement in the “Confidential to Editor” section, and submit your "Accept" recommendation.

Reviewer #1: All comments have been addressed

Reviewer #2: (No Response)

2. Is the manuscript technically sound, and do the data support the conclusions?

Reviewer #1: Yes

Reviewer #2: Partly

3. Has the statistical analysis been performed appropriately and rigorously? 

Reviewer #1: Yes

Reviewer #2: Yes

4. Have the authors made all data underlying the findings in their manuscript fully available?

Reviewer #1: No

Reviewer #2: Yes

5. Is the manuscript presented in an intelligible fashion and written in standard English?

Reviewer #1: No

Reviewer #2: Yes

6. Review Comments to the Author

Reviewer #1: The authors have responded adequately to the reviewers comments. The issued raised in the abstract, introduction, methods and discussion section have been addressed,

Reviewer #2: (No Response)

7. PLOS authors have the option to publish the peer review history of their article (what does this mean?). If published, this will include your full peer review and any attached files.

Reviewer #1: No

Reviewer #2: **Yes: **Bekele Boche Anbase

---

## [Author Response · Author response to Decision Letter 1]

12 Nov 2022

Comments for authors 

Abstract part: 

“Preventive measures taken toward preventing COVID-19 infection were decreased post-COVID-19 vaccination” this statement is not clear. Make it clear and rewrite again.

Response: Rewritten as ‘‘ COVID-19 infection preventive protocols like keeping distance, hand wash using soap, wearing mask and using sanitizer were decreased post vaccination’’.

In the result part of abstract we expected you to describe factors with its statistical analysis and significance.

Response: Comment accepted and included 

Introduction part:

Why you select Oromia region for you study and health professionals as study participants?

o Response: Oromia region is selected due administrative issues and, the reason for selecting the health professional as study participant was because of the country gave priorities of providing vaccination to the health workforce and prioritized population groups. So it was difficult to include all the general population. 

How have justified whether there was a gap and what it is?

o Response: I think the gap is mentioned in this section….ie. the following statements and others ,,, ‘‘Moreover, the rapid development of vaccines casts doubt on safety. Previously, the rapid developments of the vaccines have been linked to different adverse issues (11, 12). For example, the swine flu vaccine increased the risk of Guillain-Barre syndrome (13).’’ 

Methods:

Under Study area and period; Better to mention health-related data of the region (number of health facilities with its different levels, health professionals, covid-19 centers and…etc)

Response: Included as per the comment.

You didn’t describe or include your study population?

Not clearly described inclusion criteria of your study.

Response: I think it is clearly stated under source population

Under sample size and sampling procedures: Why did you take the p-value from Saudi Arabia, out of African content? You can use the p-value from African countries which you discussed in the discussion section. Mostly non-response rates should be 5-10%, why do you consider 15%? Need justification.

Response: The concern is accepted. Since the disease was new, we were unable to get the published studies regarding the disease during proposal development time at in Ethiopia as well in Africa. So used the Saudi Arabia study since no statistical prohibition of using abroad study result. Regarding the response rate of using 15%, by assuming that non-response will be high due to the nature of the study. Fortunately we were lucky the response rate was 97.1%

How many health facilities were selected for study?

 Response: Six Health facilities (Asella Teaching and Referral Hospital=260 participants, Bokoji Hospital=50 participants, Adama Medical Hospital College=250 participants, Mojo Hospital=50 participants, Bishoftu Hospital =100 participants, and Shashemene Referral Hospital==202 participants 

From which disciplines (professions) did health care providers have participated in the study? 

Response: All professionals= check supplementary material 1(S1)

How many health care providers are taken from each discipline or profession 

 Response: Check supplementary material 1(S1)

Please clearly describe the detailed sample size calculation and indicate how you took study participants.

o Response: I think this section is clearly stated.

Need to describe sample size calculation formula and steps 

o Response: Sample size is calculated using Epi Info software; I think no need to put the formula. 

How did you consider the participants who took the covid-19 vaccine other than Asterzenica?

o Response: They were included but the numbers are insignificant 

Under data collection procedures you have to clearly describe the following concerns.

You have to cite all sources you used to develop your tool.

o Response: Cited in the revised manuscript

How many parts do your tools have? No description of it.

o Response: The tools have eight (8) parts. 

No description of about dependent and independent variables 

o Response: The comment accepted 

What is your major outcome or primary outcome?

o Response: The major outcome is the reported side effect 

What type of items your tool has and how do you collect them from study participants?

o Response: The response is given under the comment, ‘‘How many parts do your tools have?’’

How many questions did you use to assess your objectives?

o Response: Two questions (side effets in the 1st dose or 2nd dose as Yes /No)

From where you collected your data (document, interview, self-administered questioner, observations…..?

o Response: The data were collected using interview administered

How many data collectors have participated in the study?

o Response: Twenty one (21) data collectors.

Does your study contain an observation part or checklist guided?

o Response: It did not include observation checklist, it is mentioned as limitation of the study. 

Did data collectors participate in sampling processes? 

o Response: No! the supervisors participated 

Who delivered the training for data collectors?

o Response: The investigators

On how many populations do you conduct your pretest?

From where you draw the population for the pretest 

o Response: The pretest was conducted on 48 individuals (5% of the total samples) selected from Asella Health Center(13 subjects), Eteya Health Center(1 0 subjects), Sagure Health Center(10 and Adama Health Center(15 subjects)

As you collected data by Likert scale it needs to do Cronbach's alpha to check your data's internal consistency. So, check and perform it.

Response: Thanks for your suggestion but only one question is collected using likert scale, so need to check chronbach’s alpha. The question was’

 ‘Do you agree that taking the vaccine will help to prevent the pandemic?’ (Strongly agree, Agree, Neutral, Disagree, Strongly disagree).

Results

It is good and presented in a good manner, but Socio-demographic characteristics, Chronic comorbidities conditions of participants, COVID-19 status of the participants before the vaccination, and behavioral factors you cited as S1, S2, S3, S4, what does it implies or why you didn’t put their result? 

Response: S1, 2, 3 … Is to mean supporting information. It is cited to decrease the number of tables cited to respect the submission guideline. Their result already presented in the manuscript. 

 Don’t copy all findings presented in the table or figures to the description part, only focus on the major result. 

Response: comment accepted 

Why being orthodox or other religion did significantly associate with post- covid 19 vaccination side effects? You can interpret this part? It is not likely to happen? Need to review your data again.

o Response: Religion could be a factor

Discussion and conclusion

Discussion 

You are not expected to discuss all your findings, Focus only on the major findings. 

You didn’t discuss the factors responsible for Post COVID-19 vaccination and its significance. You missed very important points and focused on minor findings.

You didn’t address about statistically significance of factors responsible for Post COVID-19 vaccination 

o Response: Dear reviewer thank you for your concern but the factors already discusses

You mentioned “The study tried to compare the events before and after the vaccine preventive measures toward COVID-19 infection” Is it you conducted a comparative study?

o Response: It is not to indicate comparative study. If you see the majority of the study showed results of 1st dose 

Conclusion 

You should describe only the major findings and factors you identified 

No need of describing all figures of your findings

Be specific and conclusive.

“In addition, further research on the cold chain of the vaccine, the reported side effect may be due to the problem related to the storage, transportation, and problem-related to the administration of the vaccine”. Did you investigate these factors?

o Response: The cold chain not investigated but some of the reported side effect may be due to improper handling of the vaccine. This is our suggestion. It could be a factor. The rest comment accepted for change.

---

## [Editor Report · Decision Letter 2]

15 Nov 2022

Post COVID-19 vaccine vaccination side effects and associated factors among vaccinated health care providers in Oromia region, Ethiopia in 2021

PONE-D-22-03691R2

Dear Dr. Tafa,

We’re pleased to inform you that your manuscript has been judged scientifically suitable for publication and will be formally accepted for publication once it meets all outstanding technical requirements.

Kind regards,

Mohd Adnan, PhD

Academic Editor

PLOS ONE

Additional Editor Comments (optional):

Manuscript is significantly improved by the authors and now can be accepted in its current form.
---

## [Editor Report · Acceptance letter]

24 Nov 2022

PONE-D-22-03691R2 

Post COVID-19 vaccination side effects and associated factors among vaccinated health care providers in Oromia region, Ethiopia in 2021 

Dear Dr. Tafa:

I'm pleased to inform you that your manuscript has been deemed suitable for publication in PLOS ONE. Congratulations! Your manuscript is now with our production department. 

Kind regards, 

on behalf of

Dr. Mohd Adnan 

Academic Editor

PLOS ONE